

# Expectation maximization—vector approximate message passing based generalized linear model for channel estimation in intelligent reflecting surface-assisted millimeter multi-user multiple-input multiple-output systems

Shoukath Ali K[1], Sajan P Philip[2], Arfat Ahmad Khan[3], Leeban Moses[2], Korhan Cengiz[4], Sedat Akleylek[5] and Nikola Ivković[6]

[1] Department of Electronics and Communication Engineering, Presidency University, Bengaluru, Karnataka, India
[2] Department of Electronics and Communication Engineering, Bannari Amman Institute of Technology, Erode, Tamil Nadu, India
[3] Department of Computer Science, College of Computing, Khon Kaen University, Khon Kaen, Thailand
[4] Department of Electrical-Electronics Engineering, Istinye University, Istanbul, Turkey
[5] Institute of Computer Science, University of Tartu, Tartu, Estonia
[6] Faculty of Organization and Informatics, University of Zagreb, Pavlinska, Varaždin, Croatia

Corresponding author
Shoukath Ali K,
shoukathali.k@presidencyuniversity.in

## ABSTRACT

Channel estimation poses a main challenge in intelligent reflecting surface (IRS)-assisted millimeter wave (mmWave) multi-user multiple-input multiple-output (MIMO) systems due to the substantial number of antennas at the base station (BS) and the passive reflective elements within the IRS lacking sufficient signal processing capabilities. This article addresses this challenge by proposing a channel estimation technique for IRS-assisted mmWave MIMO systems. The problem of channel estimation is normally taken as a compressed sensing (CS) problem, typically addressed through algorithms such as Orthogonal Matching Pursuit (OMP), Generalized Approximate Message Passing (GAMP), and Vector Approximate Message Passing with Expectation-Maximization (EM-VAMP). EM-VAMP demonstrates better performance only when a Gaussian mixture (GM) distribution is chosen as the prior for the sparse channel, especially at high signal-to-noise ratios (SNRs). To address this, the article introduces the application of generalized linear models (GLMs), extensions of standard linear models, providing increased flexibility in modeling data that deviates from Gaussian distribution. Numerical results unveil that the proposed Its EM-VAMP-GLM is much more robust to the existing OMP, GAMP and EM-LAMP algorithms.

## INTRODUCTION

During the past few years, there have been technological advancements and widespread availability of smart devices at affordable cost. These advancements cause gradual enhancements in the number of users connected to the internet. This trend experienced a significant boost in the aftermath of the pandemic, and it has caused congestion in the internet traffic, especially operating at the lower frequency bands (*Han et al., 2015*; *Raghavan et al., 2016*). To overcome such demands in the current scenario and to satisfy the needs of the future, researchers across the globe are exploring the potential of millimeter-wave (mmWave) frequency bands (*Gao et al., 2017*). However, the mmWave frequency band suffers substantial path loss. Large antenna arrays combined with mmWave technology, or massive multi-input, multi-output (MIMO) systems, are introduced to surmount this issue and ensure a higher data rate (*He et al., 2018*). Using a large antenna array comes with hardware complexity overhead in the form of dedicated radio frequency (RF) chain requirements and increased power consumption. Numerous studies have demonstrated that these effects can be mitigated using different beamforming techniques and an advanced lens antenna array system (*Yang et al., 2018*).

Intelligent reflecting surface (IRS) has been explored by many researchers because of its abilities to enhance the effectiveness of wireless communication systems. In fact, the effectiveness of wireless communication systems can further be enhanced, if we opt for the integration of IRS and the existing technologies for tackling various issues, such as channel estimation. In *Pan et al. (2022)*, the authors work on the modelling of channel under different conditions by addressing several research issues.

The both passive IRS-aided systems and active IRS-aided systems is discussed below.

### Passive IRS

A passive and low-cost reflecting element is capable for accommodating future communication systems. Reconfigurable intelligent surface (RIS)/IRS assists in handling the problem of channel estimation. In IRS, we fabricate cost effective reflective elements that are capable of reflecting signals in the desired directions. Also, IRS have low power requirements because of not requiring a dedicated hardware components like traditional RF chains, which leads to the cost reduction as well. In terms of installment, IRS can easily be installed on buildings, street lamps and ceilings (*Wu & Zhang, 2020*; *Pan et al., 2021*). However, the main issue is that it can only work to reflect the incoming signals in the desired directions without amplifying the strength of signals, which is the fundamental difference between the active and passive IRS (*Yuan et al., 2021*).

### Active IRS

Compared to passive IRS, it is able to amplify the signals due to being able to have hardware components like amplifiers or phase shifters (*Zhang et al., 2023*). Because of having additional hardware components, it consumes more power compared to passive IRS systems. In *Zhi et al. (2022)*, the researchers have discussed the active and passive IRS-aided IRS systems in a detailed way.

In massive MIMO systems, passive IRS has emerged as a relatively new approach for future-generation networks, as it may offer higher capacity, coverage, and energy efficiency (*Wu et al., 2021*). The IRS contains a large number of antennas which can be independently adjusted based on the data and algorithms. The main components of IRS are low-cost passive reflective elements consisting of a reflector that reflects the incoming signal with reconfigurable levels of angles and peaks (*Huang, Mo & Yuen, 2020*). This functionality addresses the challenges related to signal blockage, coverage issues, low signal quality and interference. Thus, joining IRS with the mmWave band is a promising method for upcoming transmission/reception systems like multi-user MIMO systems. channel state information (CSI) is crucial for optimal beamforming to realize IRS-aided massive MIMO systems. As there may be an incompatibility in the processing and sensing of pilot signals, the accumulation of CSI is extremely challenging. Each IRS consists of hundreds of larger elements, and these cascaded channels require higher pilot overhead to estimate channels. Hence, In the proposed work, passive RIS-aided mmWave MIMO system is discussed to use low power and reduce the hardware complexity of the systems.

The existing methods and techniques for IRS-aided multi-user MIMO systems need further exploration for enhanced channel estimation accuracy with minimal pilot overhead (*Jensen & Carvalho, 2020*). In *Huang et al. (2019)*, the authors discuss an optimal IRS pattern with antenna gain, demonstrating improved accuracy in contrast to the on-or-off approach discussed in *Mishra & Johansson (2019)*. However, the proposed system witnesses higher pilot overhead, which limits the performance of the IRS (*Wang et al., 2020*).

The author proposed RIS-aided massive MIMO systems that work under the scheme of two-timescale transmission. A system model with impact of electromagnetic interference (EMI) and without impact of EMI for spatially independent Rician fading channels is considered and discussed. A closed-form expression is derived for a lower bound to get an achievable rate. In *Zhi et al. (2023)*, the researchers work on the estimation of channel during the uplink transmission by applying the Linear MMSE (LMMSE) estimator. Finally, the researchers propose a low complex MRC detector.

In *Zhi et al. (2022)*, the researchers work on performance of RIS assisted massive MIMO by using ZF detector under the scenario of imperfect channel estimations. They utilize MMSE for the estimation of channel by reducing the overhead with the aim of calculating the channel. They consider the uplink transmission scenario and work on the optimization of sum rate and minimum user rate with the help of the Majorization-Minimization (MM)- based algorithm.

In *Ardah et al. (2021)*, *Fazal-E-Asim et al. (2023)*, the researchers work on proposing a non-iterative RIS assisted based estimation of channel. The proposed framework is based on two stages. They compute the directions of arrival and departures in the initial step. Following the first step, they compute the parameters, such as elevation and azimuth angles, path gains, *etc.*, in the next step. The final results unveil better performances of the proposed framework than existing methodologies.

In *Fazal-E-Asim et al. (2021)*, the researchers work on estimating a frequency selective mmWave channel by proposing a method with the help of butler matrices in analog

domain. The final results unveil that the proposed framework exhibit better performances than existing work in terms of delays, gains, and SNR.

In *Dai & Wei (2022)*, passive IRS elements are discussed for the downlink system model. Actually, it is difficult to compute the uplink and downlink channels simultaneously. Therefore, normally, researchers compute the uplink channel first, and then move towards estimating the channel during the downlink transmission. However, some researchers have worked on the estimating of channel during both transmissions in a cascaded way by using algorithms like LS (*Mishra & Johansson, 2019*), and MMSE (*Nadeem et al., 2020*). They find out that the pilot overhead gets significantly enhanced during the cascaded channel estimation because of the multiplication of number of RIS elements and antennas at BS.

In some of the works carried out in this direction, a compressive sensing (CS) scheme is used to reduce the pilot overhead by exploiting the spatial characteristics of channels, which uses classical compressive algorithms like Orthogonal Matching Pursuit (OMP) (*Liu et al., 2020*). However, the channel estimation accuracy falls short in the OMP scheme due to the cascaded and sparse nature of the channels (*Tropp & Gilbert, 2007*). Pilot overhead may be reduced using Double Structured Orthogonal Matching Pursuit (DS-OMP), which depends on cascading the computation of channels (*Dai & Wei, 2022*). In DS-OMP, different users are assigned completely non-zero rows and partially non-zero columns, and joint estimation is performed using the traditional OMP (*Wei, Shen & Dai, 2021a*, *2021b*) for channel estimation. In OMP and DS-OMP schemes, higher signal dimension increases the computational complexity and the pilot overhead. This approach, in turn, results in higher channel estimation errors. An iterative CS algorithm, Approximate Message Passing (AMP), is presented in *Lecun, Bengio & Hinton (2015)*, *Borgerding, Schniter & Rangan (2017)* to overcome the challenges mentioned above. The AMP algorithm effectively reduces the computational complexity as the signal dimension increases, but the channel estimation accuracy may still be improved.

The AMP algorithm offers enhanced performance with low computational complexity by eliminating matrix inversion (*Lecun, Bengio & Hinton, 2015*). However, the performance depends on using an independent and identically distributed (IID) sub-Gaussian measurement matrix. Even minor deviations in the IID Gaussian measurement matrix can lead to algorithm divergence. In AMP, the severe restriction of the measurement matrices and fixed shrinkage parameters limits the application for solving the CS problem (*Borgerding, Schniter & Rangan, 2017*; *Ali et al., 2023*). When applied to channel estimation in mmWave Multi-User MIMO systems, the usage of fixed shrinkage parameters in the AMP technique indirectly degrades the performance of the proposed system. This challenge may be resolved using the learned AMP (LAMP) network proposed recently (*Borgerding & Schniter, 2016*; *Wei, Hu & Dai, 2021*). LAMP is a modified version of AMP that uses a deep neural network (DNN) to optimize linear and nonlinear shrinkage parameters. In LAMP, each layer within the iteration has different shrinkage parameters. However, the LAMP network also suffers under the IID sub-Gaussian matrix, which reduces the estimation accuracy. Thus, it is worth noting that using a soft threshold shrinkage function in the LAMP network does not necessarily ensure satisfactory channel estimation accuracy.

In *Wei et al. (2019)*, the researchers discuss an AMP-based network with deep residual learning (LampResNet) for the beam space channel estimation. This work comprises the following major parts: LAMP network and deep residual learning network (ResNet). The preliminary channel computation related matrices are obtained from the first component, and ResNet is employed to reduce the noise by coarse refining. The authors point out two challenging problems that must be addressed in the LAMP network: fixed shrinkage parameter for each layer and the large memory overhead. Thus, a new approach called the hyper network-assisted recurrent LAMP (HNR-LAMP) network is introduced to resolve these challenges. However, in the scenarios involving non-IID Gaussian matrices, understanding and interpreting the parameters learned by the LAMP algorithm is challenging (*Wei et al., 2019*). As a result, the LAMP algorithm does not provide satisfactory channel estimation accuracy in IRS-assisted millimeter multi-user multiple-input multiple-output systems. Improved versions of the AMP algorithm known as VAMP and bilinear adaptive VAMP (*Rangan, Schniter & Fletcher, 2019*) have been introduced to address the limitations of LAMP in channel estimation problems. These algorithms exhibit better performance and faster convergence than conventional AMP while maintaining efficiency for a broader class of large random matrices.

The literature review suggests that the AMP-based channel estimation yields favorable outcomes in estimation accuracy in IRS-aided mmWave multi-user MIMO systems, influencing our choice to adopt AMP as a base for the proposed algorithm (*Ruan et al., 2022*). Likewise, VAMP extends the capabilities of AMP from IID Gaussian matrices to a larger class of rotationally invariant matrices. In the proposed algorithm, The VAMP algorithm with expectation-maximization (EM-VAMP) (*Schniter, Rangan & Fletcher, 2016*) is considered to avoid specifying detailed prior on channel distribution and noise variance. The investigation results show that the proposed algorithm provides better performance compared to the existing frameworks. Generalized linear models (GLMs) are extensions of the standard linear models that offer increased flexibility in modeling data that does not necessarily follow a Gaussian distribution or exhibit constant variance (*Wu & Zhang, 2018*).

This article demonstrates how EM-VAMP can be extended with GLM as EM-VAMP-GLM, a novel and robust algorithm against ill-conditioning in the measurement matrix compared to OMP, GAMP, and EM-LAMP. We consider IRS assisted Uplink mmWave multi-user MIMO systems in the article. Firstly, we formulated the signal received at the BS from the end users. As far as channel modelling is concerned, we use Saleh-Valenzuela channel model for modelling the channel between BS to IRS and IRS to end users. Following which we proposed the EM-VAMP-GLM scheme. The final results show that the proposed scheme exhibits low normalized mean square error (NMSE) compared to the existing approaches, such as OMP, GAMP, and EM-LAMP.

## SYSTEM MODEL

In this article, we consider the uplink scenario for mmWave multi-user MIMO assisted with IRS, as shown in Fig. 1. The number of users, is represented by K. We consider uniform planner array (UPA) with M antennas. IRS is equipped with N reflecting elements, where control of these elements is given to BS (*Mishra & Johansson, 2019*).

In frequency division duplex (FDD), the reciprocity of channels between the uplink and downlink channels does not hold. In this article, we consider FDD scheme, and the channel between users and BS is computed in a traditional way by switching IRS elements off. The main focus is on getting the CSI of reflecting link.

$\mathbf{G} \in \mathbf{C}^{M \times N}$ is the channel between IRS and BS, and $\mathbf{f}_k \in \mathbf{C}^{N \times 1}$ is the channel from the $k^{th}$ user $(k = 1, 2, .., K)$ to IRS.

In the uplink transmission signal model, users communicate with the BS. At BS, the received pilot signal can be written as:

$$\mathbf{y}_k = s_k \mathbf{w}_k \mathbf{G} \mathbf{\Phi} \mathbf{f}_k + n_k, \tag{1}$$

$\mathbf{y}_k$ denotes the received signal at the BS, $\mathbf{w}_k \in \mathbf{C}^{1 \times M}$ is a precoding vector, $s_k \in \mathbf{C}$ is the transmitted signal from the user, and $\mathbf{n}_k \in \mathbf{C}$ represents the additive noise. $\mathbf{\Phi} \in \mathbf{C}^{N \times N}$ represents the reflecting matrix. Assume $\mathbf{\Phi} = diag(\phi_1, \phi_2, ..., \phi_N)$ representing the diagonal reflecting matrix, and $\phi_n$ is the $n^{th}$ IRS element.

We set $\phi_n = \beta_n e^{j\psi_n}$, where $\beta_n \in [0, 1]$ and $\psi_n \in [0, 2\pi]$. The Saleh-Valenzuela channel model is utilized for the computation of $\mathbf{G}$ and $\mathbf{f}_k$. $\mathbf{G}$ can be written as:

$$\mathbf{G} = \sqrt{\frac{MN}{L_G}} \sum_{l_1}^{L_G} \alpha_{l_1}^G \mathbf{b}(\vartheta_{l_1}^{G_t}, \psi_{l_1}^{G_t}) \mathbf{a}(\vartheta_{l_1}^{G_r}, \psi_{l_1}^{G_r})^T \tag{2}$$

$L_G$ represents the total paths between IRS and BS. Similarly, we can write $\mathbf{f}_k$ as:

$$\mathbf{f}_k = \sqrt{\frac{N}{L_k}} \sum_{l_2=1}^{L_k} \alpha_{l_2}^k \mathbf{a}(\vartheta_{l_2}^k, \psi_{l_2}^k), \tag{3}$$

$L_k$ represents the total paths between uses and IRS. $\alpha_{l_2}^k$ represents the gain, $\vartheta_{l_2}^k$ represents the azimuth angle, and $\psi_{l_2}^k$ represents the elevation angle. $\mathbf{b}(\vartheta, \psi) \in \mathbf{C}^{M \times 1}$ and $\mathbf{a}(\vartheta, \psi) \in \mathbf{C}^{N \times 1}$ are the normalized array steering vector linked with BS and the IRS. For a typical $N1 \times N2 (N = N1 \times N2)$ UPA, $\mathbf{a}(\vartheta, \psi)$ is expressed as:

$$\mathbf{a}(\vartheta, \psi) = \frac{1}{\sqrt{N}} \left[ e^{-j2\pi d \cos(\psi) \mathbf{n_1}/\lambda} \right] \otimes \left[ e^{-j2\pi d \sin(\psi) \mathbf{n_2}/\lambda} \right], \tag{4}$$

$\mathbf{n}_1 = [0, 1, ..., N_1 - 1]^T$ and $\mathbf{n}_2 = [0, 1, ..., N_2 - 1]^T$, $\lambda$ and d represent the wavelength and spacing between antennas. In this article, we set $d = \frac{\lambda}{2}$.

By setting $\phi = [\phi_1, \phi_2, ..., \phi_N]$, Eq. (1) can be written as:

$$\mathbf{y}_k = s_k \mathbf{w}_k \mathbf{G} diag(\mathbf{f}_k) \phi + n_k, \tag{5}$$

$$\mathbf{y}_k = s_k \mathbf{w}_k \mathbf{H}_k \phi + n_k. \tag{6}$$

As mentioned earlier, in this article, we focus mainly on the estimation of cascaded channel by assuming the channel between end users and BS is already known. Where $\mathbf{H}_k = \mathbf{G} diag(\mathbf{f}_k)$ is the uplink cascaded channel for the $k^{th}$ user.

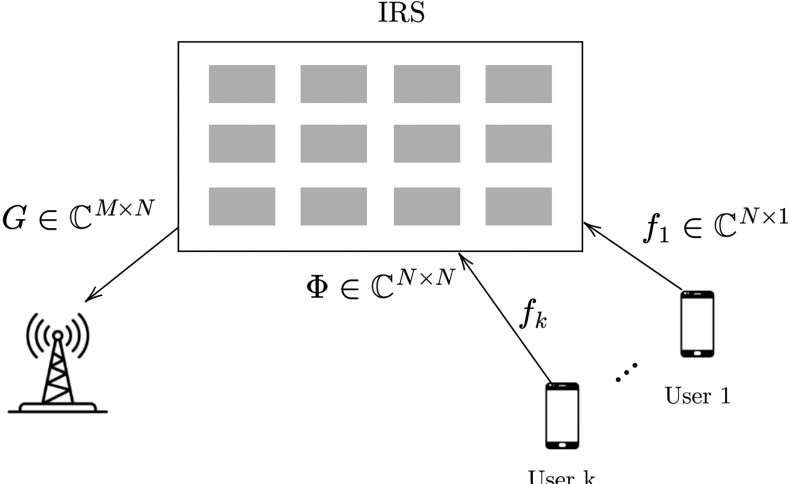

**Figure 1 The IRS assisted multi user system (*Dai & Wei, 2022*).**

The IRS has limitations in terms of signal processing; therefore, the cascaded channel computation is discussed in *Nadeem et al. (2020)*, *Zymnis, Boyd & Candès (2010)*. If we do channel estimation separately of $\mathbf{G}$ and $\mathbf{f}_k$, the computational complexity high. For reducing the computational complexity, we cascade the channel vector $\mathbf{H}_k$ and estimated using proposed method. $\mathbf{G}$ is the same for all end users because of having the common IRS between BS and users.

By decomposing $\mathbf{H}_k \in \mathbb{C}^{M \times N}$ with the help of virtual angular domain, we can write:

$$\mathbf{H}_k = \mathbf{U}_M \tilde{\mathbf{H}}_k \mathbf{U}_N^T, \tag{7}$$

where $\tilde{\mathbf{H}}_k$ represent the cascaded channel having M × N dimension. The dictionary unitary matrices are expressed as $\mathbf{U}_M$ with dimension M × M and $\mathbf{U}_N$ with dimension N × N (*Mishra & Johansson, 2019*).

Dictionary unitary matrix is a complex square matrix, which satisfy the condition $\mathbf{U}_N^T \mathbf{U}_N = \mathbf{I}$ and $\mathbf{U}_M^T \mathbf{U}_M = \mathbf{I}$.

The expression for $\mathbf{U}_M$ and $\mathbf{U}_N$ is given by

$$\mathbf{U}_M = \frac{1}{\sqrt{N}} \left[ e^{-j2\pi d \cos(\psi)\mathbf{n}_1/\lambda} \right]$$

$$\mathbf{U}_N = \frac{1}{\sqrt{N}} \left[ e^{-j2\pi d \sin(\psi)\mathbf{n}_2/\lambda} \right].$$

In wireless communication systems, particularly those involving IRS and BS, it is crucial for estimation the challenge of channel between them. By dynamically fixing phase shifts, IRS can enhance the efficiency.

In multi-user systems, the channel BS and the IRS remains constant for all users since it depends only on the determined positions.

Since different users communicate with the BS *via* the common RIS, the channel G from the IRS to the BS is common for all users.

Another solution to reduce the pilot overhead is to directly estimate the corresponding cascaded channels by utilizing the multi-user correlation. Since all users communicate with the BS *via* the same IRS, the cascaded channels associated with different users have some correlations. Thus, this multi-user correlation can be exploited to reduce the pilot overhead required by the cascaded channel estimation.

It is noted that the reflecting vector at the IRS are the same for all users.

Where $\bar{\mathbf{H}}_k$ represent the cascaded channel having $M \times N$ dimension and $\bar{\mathbf{H}}_k$ is estimated for all common users.

Apply compressive sensing techniques in $\bar{\mathbf{H}}_k$ cascade channel matrix to exploit the sparsity of the IRS-BS channels, reducing the required pilot signals for the estimation of channel.

Exploiting the common IRS-BS channel in a multi-user IRS-assisted communication system is a strategic approach to reducing channel estimation overhead.

## PROBLEM FORMULATION

The uplink cascaded channel $\mathbf{H}_k$ is calculated with the help of known pilot signals that are sent from users to the BS *via* IRS over $\mathbf{Q}$ time slots. According to Eq. (5), at the $q_{th}$ $(q = 1, 2, \ldots, \mathbf{Q})$ time slot, the received pilot signal $\mathbf{y}_{k,q}^p \in \mathbf{C}$ at the BS can be expressed as:

$$\mathbf{y}_{k,q}^p = p_{k,q}\mathbf{w}_k\mathbf{H}_k\phi_q + n_{k,q}, \tag{8}$$

where $p_{k,q}$ represents the pilot signals originated from the end users, and $\phi_q$ denotes the reflecting vector of the $q_{th}$ time slot at the IRS, and $n_{k,q}$ is the received noise on $q_{th}$ time slot at the BS that undergoes the complex Gaussian distribution that has a special characteristics of zero mean and the variance as $\sigma_n^2$.

Following the transmission of pilots at $\mathbf{Q}$ time slots, the $\mathbf{Q} \times 1$ dimensional expected overall received pilot vector $\mathbf{y}_k^p = \left[\mathbf{y}_{k,1}^p, \mathbf{y}_{k,2}^p, \ldots, \mathbf{y}_{k,Q}^p\right]^T$ by assuming $p_{k,q} = 1$ can be expressed as:

$$\mathbf{y}_k^p = \mathbf{w}_k\mathbf{H}_k\mathbf{\Theta} + \mathbf{n}_k, \tag{9}$$

where $\mathbf{\Theta} = \left[\phi_1^T, \phi_1^T, \ldots, \phi_Q^T\right]^T$ and $\mathbf{n}_k = \left[n_{k,1}, n_{k,2}, \ldots, n_{k,Q}\right]^T$.

As $vec(\mathbf{ABC}) = (\mathbf{C}^T \otimes \mathbf{A})vec(\mathbf{B})$, Eq. (9) is required to be expressed as:

$$\mathbf{y}_k^p = (\mathbf{\Theta}^T \otimes \mathbf{w}_k)vec(\mathbf{H}_k) + \mathbf{n}_k \tag{10}$$

The $vec(\mathbf{H}_k)$ is required to be expressed as $(\mathbf{U}_M^* \otimes \mathbf{U}_N)\mathbf{h}_k$, and the Eq. (10) is needed to be re-expressed as:

$$\mathbf{y}_k^p = (\mathbf{\Theta}^T \otimes \mathbf{w}_k)(\mathbf{U}_M^* \otimes \mathbf{U}_N)\mathbf{h}_k + \mathbf{n}_k. \tag{11}$$

By denoting $\mathbf{\Psi}_k = (\mathbf{\Theta}^T \otimes \mathbf{w}_k)(\mathbf{U}_M^* \otimes \mathbf{U}_N)$ and $\mathbf{h}_k \in \mathbf{C}^{MN \times 1}$, the received signal is expressed as:

$$\mathbf{y}_k^p = \mathbf{\Psi}_k\mathbf{h}_k + \mathbf{n}_k \tag{12}$$

where $\boldsymbol{\Theta}^{\mathrm{T}} \in \mathbb{C}^{\mathrm{Q} \times \mathrm{N}}$, $\mathbf{y}_k^p = [y_{k,1}, y_{k,2}, \ldots y_{k,Q}] \in \mathbf{C}^{\mathrm{Q} \times 1}$, $\boldsymbol{\Psi}_k = [\Psi_{k,1}, \Psi_{k,2}, \ldots, \Psi_{k,Q}] \in \mathbf{C}^{\mathrm{Q} \times \mathrm{MN}}$ and $\mathbf{n}_k = [n_{k,1}, n_{k,2}, \ldots n_{k,Q}] \in \mathbf{C}^{\mathrm{Q} \times 1}$.

The cascading of $\boldsymbol{\Psi}_k$ and $\mathbf{h}_k$ can be re-written as $\boldsymbol{\Psi} = [\Psi_1, \Psi_2, \ldots, \Psi_k]$ and $\mathbf{h} = [\mathbf{h}_1, \mathbf{h}_2, \ldots, \mathbf{h}_k]$ respectively.

Moreover, the pilot vector at the receiving side is required to be re-expressed as:

$$\mathbf{y}^p = \boldsymbol{\Psi}\mathbf{h} + \mathbf{n}. \tag{13}$$

To estimate the channel vector, $\mathbf{w}_k \in \mathbf{C}^{1 \times M}$ and $\boldsymbol{\Phi} \in \mathbf{C}^{\mathrm{N} \times \mathrm{N}}$ is designed as fixed value. Where $\boldsymbol{\Theta} = \left[\phi_1^T, \phi_1^T, \ldots, \phi_Q^T\right]^T \in \mathbf{C}^{\mathrm{N} \times \mathrm{Q}}$ denotes the reflecting matrix at the IRS.

Thus, similar to the pilot signals or measurement matrix, the $\boldsymbol{\Psi}_k$ is known for both the user and BS during the channel estimation. The $\mathbf{h}_k$ has to estimate based on $\mathbf{y}_k^p$ and $\boldsymbol{\Psi}_k$ for the downlink cascaded channel estimation.

## DIFFERENT TYPES OF NON IID GAUSSIAN MEASUREMENT MATRIX

For solving problems related to CS, we opt for various kinds of $\boldsymbol{\Psi}$ matrixes. In these matrixes, the correlation between columns is high. As a result, they do not exhibit satisfactory performance due to the issues linking with the constructing $\boldsymbol{\Psi}$. For exploring algorithms in terms of robustness, we have the following several kinds of matrixes:

**1) Low rank product matrix:** With the aim of constructing it, we utilize $\boldsymbol{\Psi}$ by setting the ratio $MN/N$ for analyzing in terms of robustness.

**2) Ill conditioned matrix:** With the aim of constructing it, we utilize singular matrix decomposition ($\boldsymbol{\Psi} = \mathbf{USV^T}$).

**3) Non-zero mean matrix:** All entries in $\boldsymbol{\Psi}$ follows an IID N($\mu$,1/M). For measuring the deviation, we utilize mean function.

The approaches like AMP are normally utilized for solving problems related to channel estimation. However, the issue is the minor variations from iid gaussian model leads to the diverge problems (*Rangan, Schniter & Fletcher, 2019*). Therefore, to tackle this problem, we need various kinds of $\boldsymbol{\Psi}$ in terms of evaluating robustness.

## PROPOSED CHANNEL ESTIMATION ALGORITHM

By getting inspiration from the previously outlined channel estimation framework, we can interpret the channel estimation challenge as a variant of noisy quantized compressed sensing. Various strategies have emerged to address this complexity, such as convex relaxation and GAMP (*Kamilov, Goyal & Rangan, 2012*). However, these approaches heavily rely on the assumed sparsity rate for the channel, which is typically explicitly specified or indirectly inferred through regularization terms or prior distributions. The integration of the generalized linear model (GLM) is illustrated in Fig. 2.

## PROPOSED EM-VAMP-GLM ALGORITHM

Within this section, an extension of EM-VAMP to the GLM framework is presented and mathematically derived for the analysis of the proposed methodology's performance. In

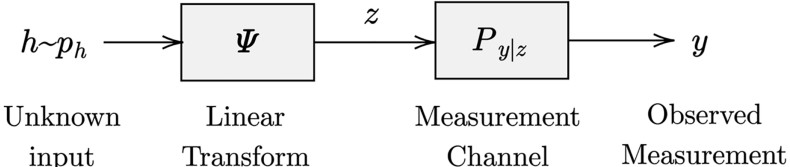

**Figure 2 Generalized linear model (GLM) (*Schniter, Rangan & Fletcher, 2016*).**

this context, the unidentified random channel vector is represented as $\mathbf{h} \in \mathbf{C}^{MN \times 1}$, observed from its noisy measurements vector denoted as $\mathbf{y} \in \mathbf{C}^{Q \times 1}$, utilizing a known measurement matrix denoted as $\boldsymbol{\Psi} \in \mathbf{C}^{Q \times MN}$. The expressed output measurements vector is formulated as:

$$\mathbf{y} = \boldsymbol{\Psi}\mathbf{h} + \mathbf{w}. \tag{14}$$

We can recover the channel vector by utilizing VAMP-GLM based technique with the help of the received measurement vector at the output side.

Here, $\mathbf{h}$ undergoes a $p(\boldsymbol{h}|\theta_{h1})$ prior density and $\mathbf{w}$ is an additive white Gaussian noise (AWGN) independent of $\mathbf{h}$ with noise variance $\boldsymbol{w} = N(0, I/\theta_{h2})$. $\boldsymbol{z}$ undergoes a $p(\boldsymbol{h}|\theta_{z1})$ prior density and noise variance $\bar{\mathbf{w}} = N(0, I/\theta_{z2})$.

The iterative AMP technique offers a less iterative solution for the aforementioned challenge, particularly when represents a $\boldsymbol{\Psi}$. However, even slight deviations from the IID sub-Gaussian model can make the AMP algorithm sensitive, leading to divergence. Contrary to AMP-based techniques like GAMP, VAMP's broader applicability in state evolution includes a wider array of matrices $\boldsymbol{\Psi}$, specifically those that possesses (RRI). $\boldsymbol{\Psi}$ can be written as $\boldsymbol{\Psi} = U_A S_A V_A^H$ by following SVD, where the elements of $V_A$ should be uniformly drawn. This essential distinction leads the VAMP approach to overcome AMP's main limitation as well as allows stability, even when dealing with ill-conditioned matrices $\boldsymbol{\Psi}$. Hence, VAMP is considered in this article.

For exhibiting its applicability to GLM with respect to small changes, we utilized the following relationship:

$$\mathbf{z} = \boldsymbol{\Psi}\mathbf{h} \quad \Leftrightarrow \quad 0 = [\boldsymbol{\Psi} - \mathbf{I}]\begin{bmatrix} \mathbf{h} \\ \mathbf{z} \end{bmatrix} \quad \Leftrightarrow \quad \bar{\mathbf{y}} = \bar{\boldsymbol{\Psi}}\bar{\mathbf{h}} + \bar{\mathbf{w}} \tag{15}$$

$$\text{where } \bar{\mathbf{y}} = 0, \bar{\boldsymbol{\Psi}} = [\boldsymbol{\Psi} - \mathbf{I}], \bar{\mathbf{h}} = \begin{bmatrix} \mathbf{h} \\ \mathbf{z} \end{bmatrix}, \text{and } \bar{\mathbf{w}} \sim N(0, I/\gamma_e) \text{ as } \gamma_e \to \infty. \tag{16}$$

Here, the two sub-vectors $\hat{\mathbf{h}}_i \in \mathbb{R}$ and $\hat{\mathbf{z}}_i \in \mathbb{R}$ is the output of $g_i$ at iteration k, and the two sub-vectors $\mathbf{r}_i \in \mathbb{R}$ and $\mathbf{p}_i \in \mathbb{R}$ is the input to $g_i$.

$\mathbf{r}_1 = \mathbf{h} + N(0, I/\gamma_1)$ denotes the pseudo-measurement model and $\mathbf{x} \sim N(\mathbf{r}_2, I/\gamma_2)$ denotes the pseudo-prior model is used to estimate the $\mathbf{h}$. Likewise, $\mathbf{p}_1 = \mathbf{z} + N(0, I/\tau_1))$ denotes the pseudo-measurements and $\mathbf{z} \sim N(\mathbf{p}_2, I/\tau_2)$ denotes the pseudo-prior is used to estimate the $\mathbf{z}$.

As we lack information about the parameters $\theta_h = (\theta_{h1}, \theta_{h2})$ and $\theta_z = (\theta_{z1}, \theta_{z2})$ for obtaining precise values of $p_h(\mathbf{h}|\theta_{h1})$ and $CN(\mathbf{y}; \mathbf{h}, \mathbf{I}/\theta_{h2})$. Similarly, we lack knowledge of parameters $\theta_z = (\theta_{z1}, \theta_{z2})$ for acquiring accurate values of $p_z(\mathbf{z}|\theta_{z1})$ and $CN(\mathbf{y}; \mathbf{\Psi h}, \mathbf{I}/\theta_{z2})$.

$\mathbf{h}_1$ is considered as the posterior probability under the prior $p(\mathbf{h}|\theta_{h1}^t)$ and message $CN(\mathbf{h}_1; \mathbf{r}_1^t, \mathbf{I}/\gamma_1^t)$, i.e., $b_{sp}(\mathbf{h}_1) \alpha p(\mathbf{h}_1|\theta_{h1}^t) CN(\mathbf{h}_1; \mathbf{r}_1^t, \mathbf{I}/\gamma_1^t)$. Hence, we can write $CN(\mathbf{h}_1, \hat{\mathbf{h}}_1, \mathbf{I}/\eta_1^t)$ where $\hat{\mathbf{h}}_1 = E[\mathbf{h}_1|b_{sp}(\mathbf{h}_1)]$ and $1/\eta_{h1}^t = \langle \text{diag}(\text{Cov}[\mathbf{h}_1|b_{sp}(\mathbf{h}_1)]) \rangle$.

In this context, we determine $\hat{\mathbf{h}}_1 = g_1(\mathbf{r}_1^t, \gamma_1)$ and $1/\eta_{h1}^t = \langle g'_1(\mathbf{r}_1^t, \gamma_1^t) \rangle / \gamma_1^t$ through the utilization of the denoising function $g_1(., \gamma)$. These values are written as follows:

$$g_{h1}(\mathbf{r}_1, \gamma_1) = \frac{\int \mathbf{h} p_h(\mathbf{h}|\theta_{h1}) N(\mathbf{h}; \mathbf{r}_1, \mathbf{I}/\gamma_1) d\mathbf{h}}{\int p_h(\mathbf{h}|\theta_{h1}) N(\mathbf{h}; \mathbf{r}_1, \mathbf{I}/\gamma_1) d\mathbf{h}}, \tag{17}$$

$$\langle g'_{h1}(\mathbf{r}_1^t, \gamma_1^t) \rangle = \frac{1}{N} \text{tr} \left\{ \frac{\partial g_i(\mathbf{r}, \gamma)}{\partial \mathbf{r}} \right\}, \text{ for i} = 1, 2. \tag{18}$$

Here, $\langle \cdot \rangle$ represents the mean coefficient value, specifically $\langle \mathbf{r} \rangle = \frac{1}{N} \sum_{i=1}^{N} \mathbf{r}_i$. In a general context, the denoising function $g_{h1}(., \gamma)$ is employed to remove noise from the pseudo-measurement $\mathbf{r}_1 = \mathbf{h} + CN(0, \mathbf{I}/\gamma_1)$ corrupted by AWGN, with prior knowledge of $p(\mathbf{h}|\theta_{h1})$.

$\mathbf{z}_1$ is considered as the posterior probability under the prior $p(\mathbf{z}|\theta_{z1}^t)$ and message $CN(\mathbf{z}_1; \mathbf{p}_1^t, \mathbf{I}/\tau_1^t)$, i.e., $b_{sp}(\mathbf{z}_1) \alpha p(\mathbf{z}_1|\theta_{z1}^t) CN(\mathbf{z}_1; \mathbf{p}_1^t, \mathbf{I}/\tau_1^t)$. Hence, we can write $CN(\mathbf{z}_1, \hat{\mathbf{z}}_1, \mathbf{I}/\eta_1^t)$ where $\hat{\mathbf{z}}_1 = E[\mathbf{z}_1|b_{sp}(\mathbf{z}_1)]$ and $1/\eta_{z1}^t = \langle \text{diag}(\text{Cov}[\mathbf{z}_1|b_{sp}(\mathbf{z}_1)]) \rangle$. Here we have $\hat{\mathbf{z}}_1 = g_1(\mathbf{p}_1^t, \tau_1)$ and $1/\eta_{z1}^t \langle g'_1(\mathbf{p}_1^t, \tau_1^t) \rangle / \tau_1^t$ with the help of noise reducing function $g_{z1}(., \tau_1^t)$:

$$g_{z1}(\mathbf{p}_1, \tau_1) = \frac{\int \mathbf{z} p_z(\mathbf{z}|\theta_{z1}) N(\mathbf{z}; \mathbf{p}_1, \mathbf{I}/\tau_1) d\mathbf{z}}{\int p_z(\mathbf{z}|\theta_{z1}) N(\mathbf{z}; \mathbf{p}_1, \mathbf{I}/\tau_1) d\mathbf{z}}, \tag{19}$$

$$\langle g'_{z1}(\mathbf{p}_1^t, \tau_1^t) \rangle = \frac{1}{N} \text{tr} \left\{ \frac{\partial g_i(\mathbf{p}, \tau)}{\partial \mathbf{p}} \right\}, \text{ for i} = 1, 2. \tag{20}$$

Here $\langle \mathbf{p} \rangle = \frac{1}{N} \sum_{i=1}^{N} \mathbf{p}_i$. Lines 16–26 of Algorithm 1 implement LMMSE estimation of $\bar{\mathbf{h}} = \begin{bmatrix} \mathbf{h} \\ \mathbf{z} \end{bmatrix}$ and the pseudo-prior

$$\bar{\mathbf{h}} = \begin{bmatrix} \mathbf{h} \\ \mathbf{z} \end{bmatrix} = N \left( \begin{bmatrix} \mathbf{r}_2 \\ \mathbf{p}_2 \end{bmatrix}, \begin{bmatrix} \mathbf{I}/\gamma_2 & \\ & \mathbf{I}/\tau_2 \end{bmatrix} \right). \tag{21}$$

Due to the gaussian nature of prior and likelihood and prior, the estimation of MAP and LMMSE is equivalent:

$$\arg \max_{\bar{\mathbf{h}}} p(\bar{\mathbf{h}}|\bar{\mathbf{y}}) = \arg \min_{\bar{\mathbf{h}}} \{ -\ln p(\bar{\mathbf{y}}|\bar{\mathbf{h}}) - \ln p(\bar{\mathbf{h}}) \} \tag{22}$$

$$\arg \min_{\mathbf{h}, \mathbf{z}} \gamma_e \|\mathbf{\Psi h} - \mathbf{z}\|_2^2 + \gamma_2 \|\mathbf{r}_2 - \mathbf{h}\|_2^2 + \tau_2 \|\mathbf{p}_2 - \mathbf{z}\|_2^2 | \theta_{h2}, \theta_{z2} \tag{23}$$

$$g_{h2}(\mathbf{r}_2, \mathbf{p}_2, \gamma_2, \tau_2, \theta_{h2}) = \mathbf{VD}(\theta_{h2} \tau_2 \mathbf{S}^T \mathbf{U}^T \mathbf{p}_2 + \gamma_2 \mathbf{V}^T \mathbf{r}_2) \tag{24}$$

---

**Algorithm 1  The EM-VAMP-GLM algorithm.**

Input: $\mathbf{\Psi} \in \mathbf{C}^{Q \times MN}$, $\mathbf{y} \in \mathbf{C}^{Q \times 1}$, T, $p_h(\boldsymbol{h}|\theta_{h1})$, $p_{y/h}(\mathbf{y}|\mathbf{h}, \theta_{h2})$, $P_z(\mathbf{z}|\theta_{z1})$ and $p_{y/z}(\boldsymbol{y}|\boldsymbol{z}, \theta_{z2})$

Define: $\mathbf{g}_{h1}(\mathbf{r}_1, \gamma_1)$, $\mathbf{g}_{h2}(\mathbf{r}_1, \gamma_1)$, $\mathbf{g}_{z1}(\mathbf{p}_1, \tau_1)$ and $\mathbf{g}_{z1}(\mathbf{p}_1, \tau_1)$ from Eqs. (17), (19), (24) and (25);

1: Initialize $\mathbf{r}_1^0, \gamma_1^0, \mathbf{p}_1^0, \tau_1^0, \theta_{h1}^0, \theta_{h2}^0, \theta_{z1}^0, \theta_{z1}^0$

2: for t = 0, 1, ..., T do

3: //Input Denoising

4: $\hat{\mathbf{h}}_1^t = g_{h1}(\mathbf{r}_1^t, \gamma_1^t, \theta_{h1})$

5: $1/\eta_{h1}^t = \langle g_{h_1}'(\mathbf{r}_1^t, \gamma_1^t, \theta h1)\rangle \gamma_1^t$

6: $\gamma_2^t = \eta_{h1}^t - \gamma_1^t$

7: $\mathbf{r}_2^t = (\eta_{h1}^t \hat{\mathbf{h}}_1^t - \gamma_1^t \mathbf{r}_1^t)/\gamma_2^t$

8: $\hat{\theta}_{h1}^{t+1} = \arg\max_{\theta_{h1}} E[\ln p_h(\mathbf{h}|\theta_{h1})|r_1, \gamma_1, \hat{\theta}_{h1}^{t+1}]$

9: //Input Denoising

10: $\hat{\mathbf{z}}_1^t = g_{z1}(\mathbf{p}_1^t, \tau_1^t, \theta_{z1})$

11: $1/\eta_{z1}^t = \langle g_{z1}'(\mathbf{P}_1^t, \tau_1^t, \theta_{z1})\rangle/\tau_1^t$

12: $\tau_2^t = \eta_{z1}^t - \tau_1^t$

13: $\mathbf{p}_2^t = (\eta_{z1}^t z_1^t - \tau_1^t \mathbf{p}_1^t)/\tau_2^t$

14: $\hat{\theta}_{z1}^{t+1} = \arg\max_{\theta_{z1}} E[\ln p_z(z|\theta_{z1})|\boldsymbol{p}_1, \tau_1, \hat{\theta}_{z1}^{t+1}]$

15: //LMMSE Denoising

16: $\hat{\mathbf{h}}_2^t = g_{h2}(\mathbf{r}_2^t, \gamma_2^t, \theta_{h2})$

17: $1/\eta_{h2}^t = \langle g_{h2}'(\mathbf{r}_2^t, \gamma_2^t, \theta_{h2})\rangle/\gamma_2^t$

18: $\gamma_1^{t+1} = \eta_{h2}^t - \gamma_2^t$

19: $\mathbf{r}_1^{t+1} = (\eta_{h2}^t \hat{\mathbf{h}}_2^t - \gamma_2^t \mathbf{r}_2^t)/(\eta_{h2}^t - \gamma_2^t)$

20: $\hat{\theta}_{h2}^{t+1} = \arg\max_{\theta_{h2}} E[\ln p_{y/h}(\mathbf{y}|\mathbf{h}|\theta_{h2})|r_2, \gamma_2, \hat{\theta}_{h2}^t]$

21: //LMMSE Denoising

22: $\hat{\mathbf{z}}_2^t = g_{z2}(\mathbf{p}_2^t, \tau_2^t, \theta_{z2})$

23: $1/\eta_{z2}^t = \langle g_{z2}'(\mathbf{p}_2^t, \tau_2^t, \theta_{z2})\rangle/\tau_2^t$

24: $\tau_1^{t+1} = \eta_{z2}^t - \tau_2^t$

25: $\mathbf{p}_1^{t+1} = (\eta_{z2}^t z_2^t - \tau_2^t \mathbf{p}_2^t)/(\eta_{z2}^t - \tau_2^t)$

26: $\hat{\theta}_{z2}^{t+1} = \arg\max_{\theta_{z2}} E[\ln p_{y/z}(\mathbf{y}|\mathbf{z}|\theta_{z2})|\mathbf{p}_2, \tau_2, , \hat{\theta}_{z2}^t]$

27: Return $\hat{\mathbf{h}}_1$

---

$$\mathbf{g}_{z2}(\mathbf{r}_2, \mathbf{p}_2, \gamma_2, \tau_2, \theta_2) = \mathbf{A}\mathbf{g}_{h2}(\mathbf{r}_2, \mathbf{p}_2, \gamma_2, \tau_2) \tag{25}$$

$$[\boldsymbol{D}]_{nn} = (\theta_{h2}\tau_2 s_n^2 + \gamma_2)^{-1} \tag{26}$$

$$\mathbf{g}_{h2}(\mathbf{r}_2, \mathbf{p}_2, \gamma_2, \tau_2, \theta_{h2}) = \mathbf{r}_2 + \mathbf{V}\mathbf{S}^T \left(\theta_{h2}\frac{\gamma_2}{\tau_2}\mathbf{I} + \mathbf{S}\mathbf{S}^T\right)^{-1}(\mathbf{U}^T \mathbf{p}_2 - \mathbf{S}\mathbf{V}^T \mathbf{r}_2) \tag{27}$$

$$\partial \mathbf{g}_{h2}/\partial \mathbf{r}_2 = \gamma_2 \mathbf{V}\mathbf{D}\mathbf{V}^T \tag{28}$$

$$\alpha_2 = \langle \mathbf{g}'_{h2}(\mathbf{r}_2, \mathbf{p}_2, \gamma_2, \tau_2, \theta_{h2}) \rangle = \frac{1}{MN} \sum_{n=1}^{MN} \frac{\gamma_2}{(\theta_{h2}\tau_2 s_n^2 + \gamma_2)} \tag{29}$$

$$\partial \mathbf{g}_{z2}/\partial \mathbf{p}_2 = \tau_2 \mathbf{S} \mathbf{D} \mathbf{S}^{\mathrm{T}} \tag{30}$$

$$\beta_2 = \langle g'_{z2}(\mathbf{r}_2, \mathbf{p}_2, \gamma_2, \tau_2, \theta_{z2}) \rangle = \frac{1}{M} \sum_{n=1}^{N} \frac{\tau_2 s_n^2}{(\theta_{z2}\tau_2 s_n^2 + \gamma_2)} = \frac{M}{N}(1 - \alpha_2). \tag{31}$$

## CHOICE OF PRIOR DISTRIBUTION

To effectively utilize the EM-VAMP in tackling the estimation challenge assisted by IRS, we choose to embrace an approximating prior family. This family presents two viable choices: a BG distribution or GM distribution, each involving unspecified parameters $\theta_{h1}$. This implies $h_n$ of $\mathbf{h}$ is taken out with the help of the subsequent equations:

$$\text{BG}: \ p(h_n|\theta_{h1}) = \lambda_0 \delta(h_n) + (1 - \lambda_0)CN(h_n; 0, \tau) \forall i \tag{32}$$

$$\text{BG}: \ p(z_n|\theta_{z1}) = \lambda_0 \delta(z_n) + (1 - \lambda_0)CN(z_n; 0, \tau) \forall i \tag{33}$$

$$\text{GM}: \ p(h_n|\theta_{h1}) = \lambda_0 \delta(h_n) + \sum_{i}^{L} \lambda_i CN(h_n; \mu_i, \tau_i) \forall i \tag{34}$$

$$\text{GM}: \ p(z_n|\theta_{z1}) = \lambda_0 \delta(z_n) + \sum_{i}^{L} \lambda_i CN(z_n; \mu_i, \tau_i) \forall i \tag{35}$$

where $\lambda_0 = \text{prob}\{h_n = 0\}$ and $\lambda_0 = \text{prob}\{z_n = 0\}$, L is the number of Gaussian distribution mixed in GM; $\lambda_i$, $\mu_i$, and $\tau_i$ are the weights, means, and variances. The channel coefficient follow the BG distributions for Eqs. (32) and (33), and GM distributions for Eqs. (34) and (35) respectively with unknown parameters $\theta_{h1}$ and $\theta_{z1}$ and noise variance $\theta_{h2}$ and $\theta_{z2}$, respectively.

The computational complexity and the performance comparison analysis is shown in Table 1.

## RESULTS AND DISCUSSION

The simulation results for the proposed EM-VAMP-GLM is discussed and compared with the existing OMP, GAMP and EM-LAMP. The simulation parameter is shown in Table 2.

The azimuth/elevation angles are uniformly generated from $(-\pi/2, \pi/2)$ along the pre-discretized virtual angle grids. Equation (3) is utilized for generating the channel $\mathbf{f}_k$. The operating frequency is considered as 28 GHz.

Where the normalized mean square error (NMSE) is defined as

$$\text{NMSE} = \frac{\text{E}\left\{ \sum_{k=1}^{K} \left\| \hat{\mathbf{h}}_k - \mathbf{h}_k \right\|_2^2 \right\}}{\text{E}\left\{ \sum_{k=1}^{K} \|\mathbf{h}_k\|_2^2 \right\}}.$$

As shown in Fig. 3, the NMSE with respect to SNR is shown between the proposed EM-VAMP-GLM algorithm and alternative approaches including OMP, GAMP, and

**Table 1 Computational complexity and performance comparison.**

| Sl No. | Name of the algorithm | Computational complexity | Comparative analysis |
|---|---|---|---|
| 1. | OMP | $O(SMN) + O(S^3N)$ | The computational complexity is high for high dimensional sensing matrix. |
| 2. | GAMP | $O(TMN)$ | The computational complexity is lower than the OMP scheme, because GAMP uses a simple matrix multiplication process. |
| 3. | EM-LAMP | $O(T(MN + N^2))$ | The computational complexity is higher than GAMP, because the incorporation of the EM concept with LAMP network improves the channel estimation accuracy. |
| 4. | EM-VAMP – GLM | $O(T(MN + N^2))$ | The Computational complexity of EM-VAMP–GLM is same with EM-LAMP. EM-VAMP–GLM is deal with vector based prior information and EM-LAMP is deal with scalar based prior information. EM-VAMP-GLM outperforms EM-LAMP and it provide more accurate prior information, better convergence during high-dimensional data. The proposed method can exploit the vectorized structure to process and interpret the data more efficiently. |

**Table 2 Simulation parameters.**

| Parameters | Value |
|---|---|
| N | 256 |
| M | 64,128,256 |
| $L_G$ | 5 |
| $L_K$ | 8 |

EM-LAMP. The results show that GAMP and EM-LAMP algorithms exhibit superior performance compared to the OMP algorithm. In the case of an IID Gaussian matrix $\Psi$, both GAMP and EM-LAMP techniques do not perform well when compared to the proposed EM-VAMP-GLM algorithm.

Figure 4 provides a comprehensive exploration of the NMSE against SNR for various shrinkage function. The performance analysis of various shrinkage functions such as GM, BG, and soft threshold shrinkage function is applied in the proposed EM-VAMP-GLM and EM-LAMP network. The proposed EM-GM-VAMP-GLM, EM-BG-VAMP-GLM and EM-ST-VAMP algorithms, is compared to the existing EM-GM-LAMP, EM-BG-LAMP and EM-ST-LAMP.

LAMP behaves certainly with IID Gaussian matrix $\Psi$, but less certainly with non-IID-Gaussian $\Psi$. To address the challenge of non-IID-Gaussian signal estimation, we introduce the EM-GM-VAMP-GLM and EM-BG-VAMP-GLM methodologies. In Fig. 4, the usage of BG or GM shrinkage function for proposed algorithm achieve better performance compared to the soft threshold shrinkage function based EM-VALP-GLM algorithm. Even when $\Psi$ is not IID Gaussian, EM-GM-VAMP-GLM and EM-BG-VAMP-GLM algorithm perform well compared to the LAMP and GAMP.

Figure 5 illustrates the relationship between NMSE and the number of iterations for the proposed EM-BG-VAMP-GLM and EM-GM-VAMP-GLM techniques across varying SNRs. Evidently, both the EM-BG-VAMP-GLM and EM-GM-VAMP-GLM algorithms exhibit convergence within a modest span of fewer than 22 iterations. This observation highlights their fast convergence rate, and reduces the computational complexity for BG
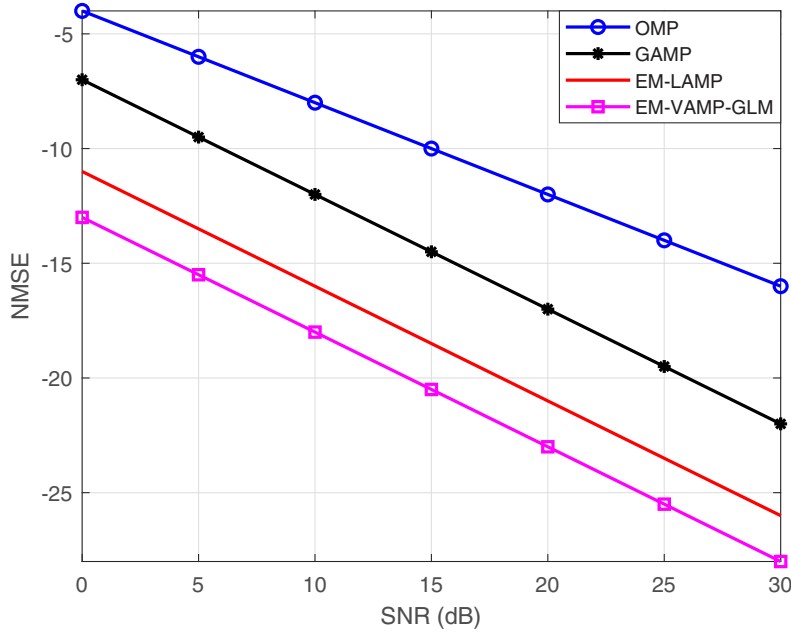

**Figure 3 NMSE *vs*. SNR.**

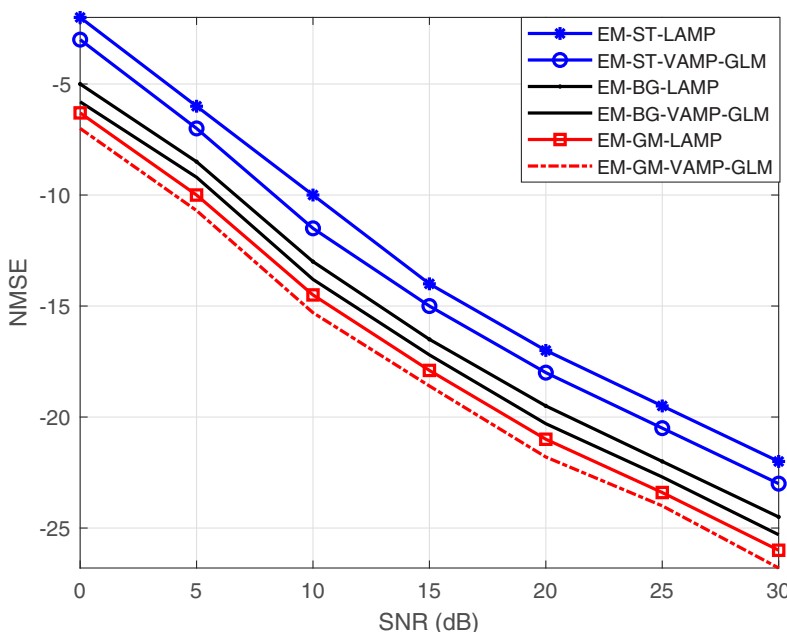

**Figure 4 NMSE *vs*. SNR with different shrinkage functions such as soft threshold, Bernoulli-Gaussian and Gaussian mixture.**

and GM shrinkage function scenarios. Notably, in both algorithms, an increase in SNR corresponds to an extended the convergence rate with a greater number of iterations, because the BG and GM shrinkage function need additional parameters to be learned in the EM update process of EM-BG-VAMP-GLM and EM-GM-VAMP-GLM algorithms.

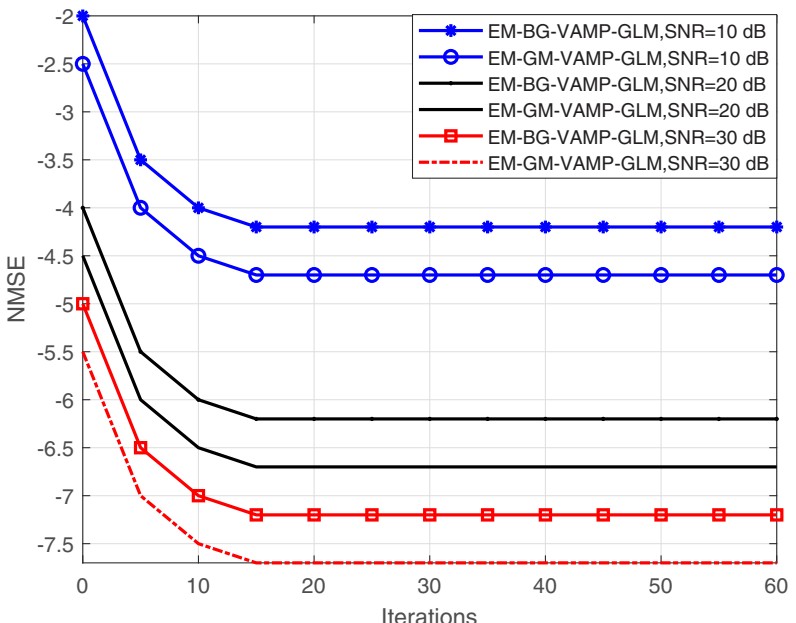

**Figure 5** NMSE *vs.* Iterations.

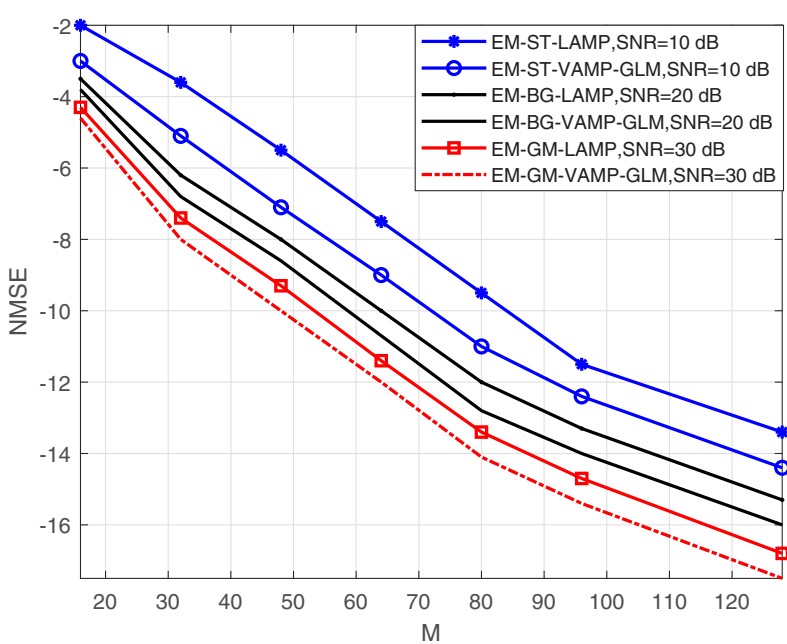

**Figure 6** NMSE *vs.* M with different SNR value.

The investigation into NMSE performance *vs.* the number of BS antennas, denoted as M for both the proposed EM-BG-VAMP-GLM and EM-GM-VAMP-GLM algorithms at different value of SNRs is shown in Fig. 6. For the increased number of antennas M, it is observed that both EM-BG-VAMP-GLM and EM-GM-VAMP-GLM algorithms consistently exhibit reduced NMSE values. The higher SNRs value provide low NMSE value for proposed EM-BG-VAMP-GLM and EM-GM-VAMP-GLM. Under sparser

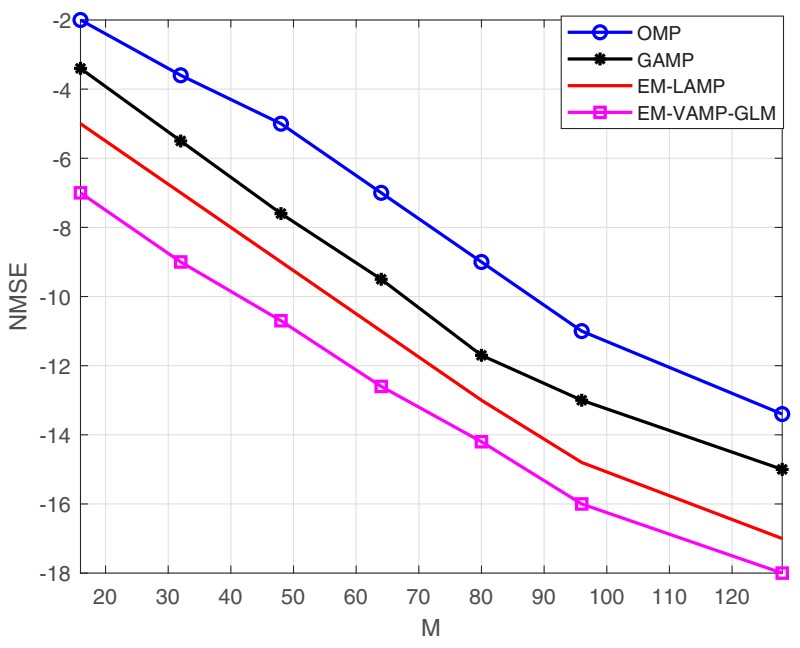

**Figure 7** NMSE *vs.* M.

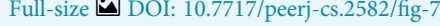

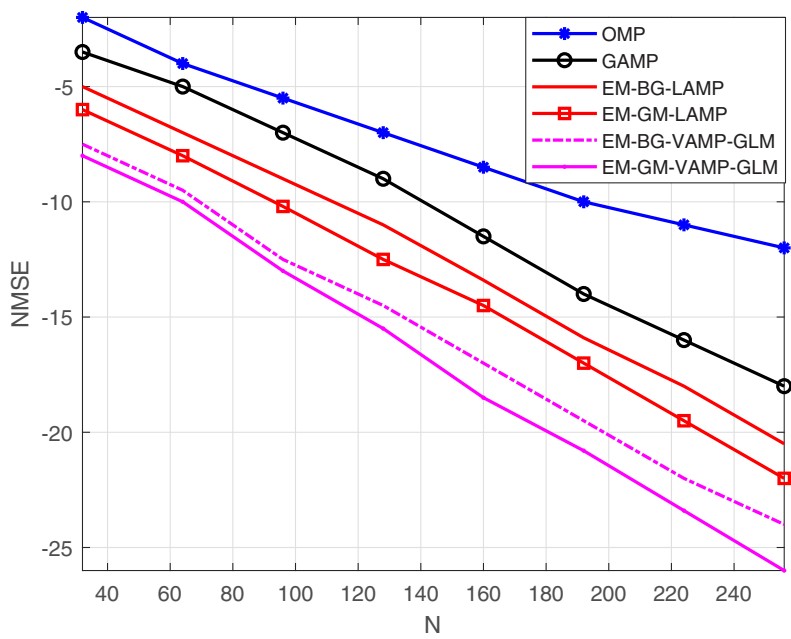

**Figure 8** NMSE *vs.* N.

environment, the scattering paths is fixed, so that the number of large non-zero elements is also fixed. The increased number of antennas provide the estimation outcomes are inherently enhanced, under the sparser conditions. Ultimately, these results highlight the practicality and appropriateness of utilizing the suggested EM-BG-VAMP-GLM and EM-GM-VAMP-GLM algorithms for efficient channel estimation with the assistance of

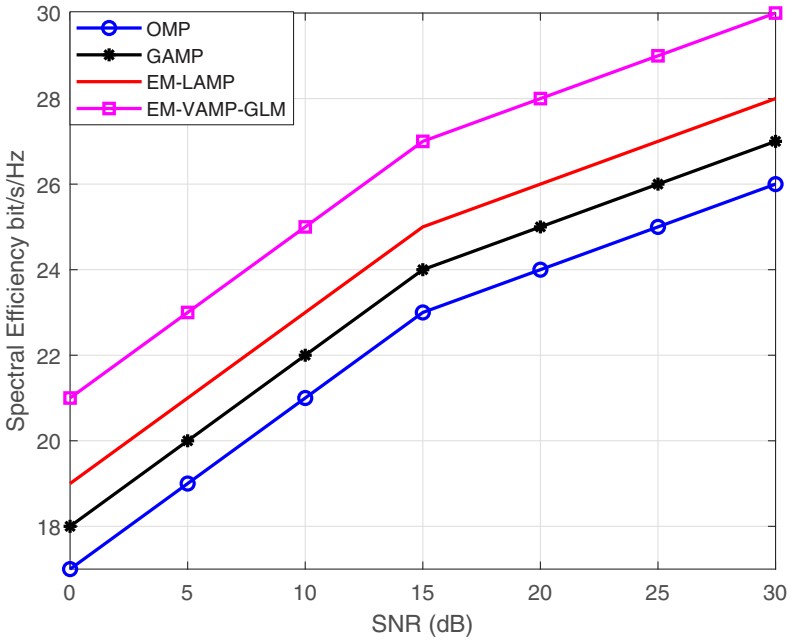

**Figure 9** SNR *vs.* spectral efficiency bits/s/Hz. 

**Table 3** SNR *vs.* spectral efficiency bits/s/Hz.

| Sl No. | SNR (dB) | OMP | GAMP | EM-LAMP | EM-VAMP–GLM |
|--------|----------|-----|------|---------|-------------|
| 1. | 0 | 17 | 18 | 19 | 21 |
| 2. | 5 | 19 | 20 | 21 | 23 |
| 3. | 10 | 21 | 22 | 23 | 25 |
| 4. | 15 | 23 | 24 | 25 | 27 |
| 5. | 20 | 24 | 25 | 26 | 28 |
| 6. | 25 | 25 | 26 | 27 | 29 |
| 7. | 30 | 26 | 27 | 28 | 30 |

IRS in mmWave communication systems. This leverages the benefits derived from angular domain sparsity.

In Fig. 7, we perform the comparison between M and NMSE. As it can be seen in Fig. 7, the proposed approach outscores the existing approaches by exhibiting less NMSE with respect to the increment in the number of antennas.

Figure 8 depicts the NMSE against the number of IRS elements, denoted as N, for the EM-BG-VAMP-GLM and EM-GM-VAMP-GLM algorithms. The quantity of IRS elements, N, increases, with both the EM-BG-VAMP-GLM and EM-GM-VAMP-GLM algorithms consistently demonstrating lower NMSE compared to the other techniques.

Figure 9 provides the SNR against the Spectral Efficiency for the proposed EM-VAMP-GLM and Existing algorithm such as OMP, GAMP and EM-LAMP algorithm. At 15 dB SNR, the proposed EM-VAMP-GLM algorithm provide 14.8%, 11% and 7%, and 8.5% of

enhancement in spectral efficiency compared to OMP, GAMP and EM-LAMP algorithm respectively. The SNRVs Spectral Efficiency is shown in Table 3.

## CONCLUSIONS

This research focuses on addressing the channel estimation problem within an IRS-assisted mmWave MIMO system. We present the EM-BG-VAMP-GLM and EM-GM-VAMP-GLM techniques for effectively computing channel vector in this scenario. By utilizing EM-VAMP-GLM, we achieve precise channel estimation without the need for crucial parameters like noise variance and prior distribution parameters. The investigation considers two potential distributions–BG and GM distributions–to identify the most suitable model for characterizing the sparse angular domain channel. We perform the comparison between EM-GM-VAMP-GLM and EM-BG-VAMP-GLM techniques against existing methods such as LAMP, GAMP, and OMP. The experimental results indicate that, especially at higher SNRs, the EM-GM-VAMP-GLM demonstrates superior computation accuracy with rapid convergence.

### Funding
The authors received no funding for this work.

### Competing Interests
Sedat Akleylek is an Academic Editor for PeerJ. The authors declare that they have no competing interests.

### Author Contributions
- Shoukath Ali K. conceived and designed the experiments, performed the experiments, analyzed the data, performed the computation work, prepared figures and/or tables, authored or reviewed drafts of the article, and approved the final draft.
- Sajan P Philip conceived and designed the experiments, performed the experiments, performed the computation work, authored or reviewed drafts of the article, and approved the final draft.
- Arfat Ahmad Khan conceived and designed the experiments, analyzed the data, performed the computation work, authored or reviewed drafts of the article, and approved the final draft.
- Leeban Moses performed the experiments, authored or reviewed drafts of the article, and approved the final draft.
- Korhan Cengiz performed the experiments, prepared figures and/or tables, and approved the final draft.
- Sedat Akleylek analyzed the data, performed the computation work, prepared figures and/or tables, and approved the final draft.
- Nikola Ivković analyzed the data, prepared figures and/or tables, and approved the final draft.

## Data Availability

The data and code for channel and measurement matrix are available in the Supplemental Files.

## Supplemental Information

Supplemental information for this article can be found online at http://dx.doi.org/10.7717/peerj-cs.2582#supplemental-information.

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
