# Peer review of "Expectation maximization—vector approximate message passing based generalized linear model for channel estimation in intelligent reflecting surface-assisted millimeter multi-user multiple-input multiple-output systems"

_PeerJ Computer Science, doi:10.7717/peerj-cs.2582_

## Round 0.1 · original submission · Major Revisions

Please see enclosed the reviewers' comments. Given the review results, I would be happy to provide opportunities to address the comments and revise the manuscript.

Reviewer 1 ·

Basic reporting

The presentation needs to be fully polished. Typos about formulas need to be corrected. The readability should be improved.

Experimental design

How to design the phase shifts of the IRS needs to be clarified.

Validity of the findings

The details about devarition from Gaussian distribution need to be highlighted and improved.

Additional comments

The authors aimed to propose an effective channel estimation algorithm for IRS-aided mmWave MIMO systems in the presence of non-Gaussian distributed variables. The proposed EM-VAMP-GLM algorithm sounds interesting, however, the paper needs improvement before accepted.
1. There exist a lot of typos and incorrect forms for the variables throughout the paper. The whole manuscript needs to be fully polished and correct the many grammar errors. Meanwhile, the errors for many variables and in many formulations should be corrected. For example, there are wrong forms of n1 and n2 in (4), and n_k in (9) should be bold form. Besides, some variables are not defined well, such as y_k which is related with symbol k. Thus, how to estimate channels for all K users? \theta in line 240 is not defined either. Please check all formulas in the paper carefully.
2. What is the purpose and impact of considering FDD scheme? Does it change anything in the algorithm design?
3. In Line 180, the authors claimed H_k is estimated separately. However, it seems that the most of algorithms estimate H_k as a whole rather than separately estimating G and f_k. Please further justify.
4. How to pre-design w_k and Phi? This needs to be clarified. What are they designed in the simulation?
5. In Line 213, the authors said \Psi_k is known. However, how to know U_M and U_N inside \Psi_k? Please justify.
6. It is hard to read the section of the proposed algorithm. Please improve its readability. Also, please highlight the difference and the ability of the proposed algorithm for tacking the non-Gaussian distribution.
7. The authors lacked introducing a promising channel estimation method proposed for IRS-aided massive MIMO systems, i.e., two-timescale design which only estimates the effective aggregated channel with very low pilot length [R1-R2]. Please add it to the Introduction. [R1] Two-timescale design for reconfigurable intelligent surface-aided massive MIMO systems with imperfect CSI [R2] Is RIS-aided massive MIMO promising with ZF detectors and imperfect CSI?

Reviewer 2 ·

Basic reporting

The notations in the manuscript are at a mess, and the English presentation is poor. The quality of the manuscript should be improved.

Experimental design

As channel estimation for IRS-assisted mmWave multi-user MIMO system has been studied before. Performance comparison with the existing works should be provided.

Validity of the findings

1. Why the direct link is ignored in the considered system?
2. As multiple users are considered, why the common IRS-BS channel is not exploited to reduce the channel estimation overhead?

Reviewer 3 ·

Basic reporting

This research focuses on addressing the channel estimation challenge within an IRS-assisted
MIMO multi-user mmWave communication system. They present the EM-BG-VAMP-GLM and EM-GM-VAMP-GLM techniques for effectively computing channel vector in this scenario.

Experimental design

This paper performs extensive experimental design to verify the effectiveness of the proposed scheme. The results are good and impressive.

Validity of the findings

The results of this paper are interesting and useful. The proposed scheme is useful.

Additional comments

The reviewer mainly has the following concerns:
1) In fact, this paper only considered one user. Hence, the authors need to remove the mult-user term. For multiuser case, the properties of this system should be used. Pls refer to the following paper and ack this work in your paper since it is closely related to this work: Channel estimation for RIS-aided multiuser millimeter-wave systems
2) It is interesting to see how this algorithm works in the case of multiusers when exploiting the special properties of the RIS-aided mmWave systems.
3) For mmWvae systems, the BS is generally equipped with hybrid precoding structure. However this algorithm works in this case?
4) The introduction is not complete. To help readers understand the recent advace in the area of RIS, the authors need to ack the following classical tutorial paper: An overview of signal processing techniques for RIS/IRS-aided wireless systems
5) In this paper, the authors mainly considered the passive RIS, which is well known to suffer from the double path fading effect. To address this issue, the researches proposed the novel concept of active RIS, which is equipped with amplifers. Pls refer to the following paper for more details: Active RIS versus passive RIS: Which is superior with the same power budget? Pls explain whether the proposed channel estimation scheme can be applicable to this case or not.

Reviewer 4 ·

Basic reporting

The paper is about channel estimation for multiuser IRS-assisted systems. The
proposed approach, named EM-VAMP-GLM (Generalized Approximate Message Passing, and Vector Approximate Message Passing with Expectation-Maximization).

Experimental design

The system model looks impractical as the IRS phase shifts could be optimized for one user at a time. Experimentally, how the authors will serve K users with one IRS is still confusing.

Validity of the findings

The results seem to be OK, but lack comparison with some stat-of-the-art comparisons.

Additional comments

The system model and the equations could be mentioned more clearly. The author could take help from the following paper, regarding writing the equations:

1- Fazal-E-Asim, F. Antreich, C. C. Cavalcante, A. L. F. de Almeida, and J. A. Nossek, “Two-dimensional channel parameter estimation for millimeter-wave systems using Butler matrices,” IEEE Trans. Wirel.
Commun, vol. 20, no. 4, pp. 2670–2684, 2021.

The authors must write the simulation parameters in a table, and also compare with the algorithms of the following papers:

2- K. Ardah, S. Gherekhloo, A. L. F. de Almeida, and M. Haardt, “TRICE:
A channel estimation framework for RIS-aided millimeter-wave MIMO
systems,” IEEE Sig. Proc. Lett, vol. 28, pp. 513–517, 2021.

3- FAsim, "Two-dimensional channel parameter estimation for irs-assisted networks", https://arxiv.org/pdf/2305.04393, 2023.

---

## Round 0.2 · Major Revisions

I offer the authors the opportunity to address critical comments from the reviewers, particularly on the aspect of novelty. Please consider the novelty carefully when revising and resubmitting the paper.

Reviewer 1 ·

Basic reporting

no comment

Experimental design

no comment

Validity of the findings

no comment

Additional comments

Thanks for the revision. No further comments.

Reviewer 4 ·

Basic reporting

The paper is about channel estimation in multiuser IRS-assisted MIMO systems. However, the proposed algorithm is very limitedly novel. Most of these compressed sensing algorithms (OMP) are very well-known in the literature. The authors must clearly mention their contribution to the state-of-the-art methods that use compressed seeing approach.

Experimental design

The main problem in the scenario is how an IRS will simultaneously serve many users at a time, especially when considered passive IRS.

Validity of the findings

The paper may be somewhat novel if multiantenna users are considered, assuming active IRS, or may be time division multiplexing among different users.

Additional comments

1- The system and channel model should be explained clearly, and the organization of equations should be revised.

2- Which codebook is used by authors for compressed sensing?

3- Compressed sensing methods are better for spectral efficiency. Therefore, a table should be given where the spectral efficiency of the proposed methods is compared with competing methods.

4- If expectation maximization (EM) is used, the maximum likelihood estimator is somehow deployed. Comparisons with some of the following work may evaluate the proposed algorithm's performance.

[R1] F. Asim, Josef Nossek,“Two-dimensional channel parameter estimation for
millimeter-wave systems using Butler matrices,” In: IEEE Transactions on Wireless
Communications ( Volume: 20, Issue: 4, April 2021).

[R2] K. Ardah, S. Gherekhloo, A. L. F. de Almeida, and M. Haardt, “Trice:
A channel estimation framework for RIS-aided millimeter-wave MIMO
systems,” IEEE Sig. Proc. Lett, vol. 28, pp. 513–517, 2021.

---

## Round 0.3 · accepted · Accept

The revision appears to the level that is satisfactory to the reviewers.

Reviewer 4 ·

Basic reporting

'no comment'

Experimental design

'no comment'

Validity of the findings

'no comment'

Additional comments

My comments are incorporated. I do not have further comments.